# Histological Evaluation of Cassava Starch/Chicken Gelatin Membranes

**DOI:** 10.3390/polym14183849

**Published:** 2022-09-14

**Authors:** Carlos Humberto Valencia-Llano, Jorge Iván Castro, Marcela Saavedra, Paula A. Zapata, Diana Paola Navia-Porras, Edwin Flórez-López, Carolina Caicedo, Heidy Lorena Calambas, Carlos David Grande-Tovar

**Affiliations:** 1Research Group in Biomateriales Dentales, School of Odontología, Faculty of Health, Campus San Fernando, Universidad del Valle, Calle 4B # 36-00, Cali 76001, Colombia; 2Research Group SIMERQO, Department of Chemistry, Faculty of Natural and Exact Sciences, Campus Melendez, Universidad del Valle, Calle 13 No. 100-00, Santiago de Cali 76001, Colombia; 3Research Group of Polímeros, Department of Chemistry, Faculty de Chemistry and Biology, Universidad de Santiago de Chile, USACH, Santiago 9170020, Chile; 4Research Group Biotecnología, Faculty of Engineering, Universidad de San Buenaventura Cali, Carrera 122 # 6-65, Santiago de Cali 76001, Colombia; 5Research Group in Química y Biotecnología QUIBIO, Universidad Santiago de Cali, Calle 5 No 62-00, Cali 760035, Colombia; 6Research Group GIGAE3D, Faculty of Engineering, Unidad Central del Valle del Cauca (UCEVA), Carrera 17ª 48-144, Tuluá 763022, Colombia; 7Research Group in Desarrollo de Materiales y Productos, Centro Nacional de Asistencia Técnica a la Industria (ASTIN), SENA, Cali 760003, Colombia; 8Research Group of Fotoquímica y Fotobiología, Universidad del Atlántico, Carrera 30 Número 8-49, Puerto Colombia 081008, Colombia

**Keywords:** biocompatibility, biocomposite, cassava starch, chicken gelatin, composite membranes, tissue engineering

## Abstract

The use of biopolymers for tissue engineering has recently gained attention due to the need for safer and highly compatible materials. Starch is one of the most used biopolymers for membrane preparation. However, incorporating other polymers into starch membranes introduces improvements, such as better thermal and mechanical resistance and increased water affinity, as we reported in our previous work. There are few reports in the literature on the biocompatibility of starch/chicken gelatin composites. We assessed the in vivo biocompatibility of the five composites (T1–T5) cassava starch/gelatin membranes with subdermal implantations in biomodels at 30, 60, and 90 days. The FT-IR spectroscopy analysis demonstrated the main functional groups for starch and chicken gelatin. At the same time, the thermal study exhibited an increase in thermal resistance for T3 and T4, with a remaining mass (~15 wt.%) at 800 °C. The microstructure analysis for the T2–T4 demonstrated evident roughness changes with porosity presence due to starch and gelatin mixture. The decrease in the starch content in the composites also decreased the gelatinization heats for T3 and T4 (195.67, 196.40 J/g, respectively). Finally, the implantation results demonstrated that the formulations exhibited differences in the degradation and resorption capacities according to the starch content, which is easily degraded by amylases. However, the histological results showed that the samples demonstrated almost complete reabsorption without a severe immune response, indicating a high in vivo biocompatibility. These results show that the cassava starch/chicken gelatin composites are promising membrane materials for tissue engineering applications.

## 1. Introduction

Biocomposites have recently gained much attention due to the global risk of indiscriminate plastic use with devastating consequences for the living planet and human health [1]. Many researchers have worked on developing new materials that contribute to a circular economy, in which the use of single-use plastics is reduced to increase the use of materials from natural sources, whose residues can be re-incorporated into other processes or composted [2]. Biocomposites are well received in medicine for developing hydrogels, sutures, patches, and membranes for cell regeneration, avoiding rejection problems, severe immune responses, and other common complications among patients who use implants for recovery from injured tissues [3,4,5].

A polymer biocomposite contains a polymer matrix, and one or more dispersed fibers introduce new properties to the polymer matrix [6,7]. The mixture includes several biodegradable components, improving compatibility and reducing the long-term contamination [8]. One of the essential attributes of biocomposites in biomedical applications is that they encourage cell regeneration through the exchange of nutrients, the removal of waste, and cell adhesion so that the tissue can regenerate [9]. Biocomposites then must be porous biocompatible materials that allow cell adhesion and proliferation.

Starch is a polymer found in all plants with seeds but in more significant proportion in tubers and grains. Starch is an amylose and amylopectin-derived polymer [10]. Amylose is a linear polymer consisting of D-glucose α-(1,4) glycosidic bonds [11]. The actual structure of amylose is helical due to the backbone’s α-(1,4) glycosidic bonds. Amylopectin is a highly branched glucose whose system is based on shorter chains than amylose with α-(1,4) glycosidic bonds crosslinked with α-(1,6)-glycosidic bonds [12].

Interest in the development of starch-derived materials has increased due to their extraordinary versatility mediated by their inherent properties as a polysaccharide (hydrophilicity, membrane-forming capacity, biodegradability, and enzymatic degradability) [13,14]. Other incentives for the use of starch arise from the easy preparation, abundance, and high performance in drug delivery [15,16], tissue engineering [17], antimicrobial agents [18], and diagnostic images [19]. However, using pure starch brings insolubility, especially in cold water, composition variations, source-depending properties, and the possibility of structure breaking under overheating. For this reason, chemical, physical and enzymatic modifications of native starch have been proposed to improve its properties and applications [14]. On the other hand, the Preparation of starch composites with other polymers to enhance their properties has also been used for biomedical applications [20] and in the bone tissue regeneration [21].

Gelatin is obtained from the hydrolytic cleavage of hydrogen and covalent bonds in collagen with a random coil structure [22]. This protein is obtained mainly from the connective tissue of cattle, pigs, and fish [23]. It is applied extensively in the food [24], cosmetics, and pharmaceutical industries [25], thanks to its viscoelastic properties and its ability to form gels [17]. Gelatin has been used in biomedical applications [26] based on biocompatibility, biodegradability, and many functional groups. These properties are responsible for easy chemical modification in the hydrogel preparation and other valuable devices for the cell regeneration [3,27,28,29] and wound dressing [30]. It has been shown that when gelatin is used in bandages, it extraordinarily favors cell development due to its remarkable similarity with the extracellular matrix [31]. However, this material is brittle and fragile, especially for applications in complex tissue regeneration, requiring functionalization or mixing with other materials to improve mechanical properties [32,33].

In this regard, generating different mixtures between gelatin and various components would improve the physical conditions in a biological environment. The improvement would involve the formation of scaffolds for skin lesions or as barriers in processes where it is required to promote bone healing, such as the guided bone regeneration technique. The skin is a vital organ with many functions to protect the tissues; when part of it is lost due to accidents or burns, it is essential to protect the injured tissue by employing scaffolds to prevent water loss and infection. The scaffolds can be natural or synthetic and must comply with the properties of being biocompatible, stable for the healing duration, absorbent of tissue exudates, and ideally with the ability to bio-integrated allowing the closure of the wound properly and be bacteriostatic [34].

In guided bone regeneration techniques, the aim is to stimulate tissue neoformation by using bone substitutes and placing a membrane between the graft material and the soft tissue to act as a barrier that prevents the smooth tissue cells from colonizing the graft site. This barrier must be made of a biocompatible material, one of the most commonly used collagen [35].

We published the synthesis and physicochemical characterization of five different formulations of membrane cassava starch and chicken gelatin composites for food packaging applications [36]. However, despite the recorded applications of cassava starch with gelatin membranes for food packaging, these membranes’ biocompatibility has not been previously reported. Therefore, we report the study of in vivo biocompatibility during 90 days of implantation of five different formulations of cassava starch and chicken gelatin biocomposites. The results show that this type of membrane is beneficial for regenerating subdermal tissue, thanks to its rapid reabsorption in the tissues without generating an abnormal immune response in the host.

## 2. Materials and Methods

### 2.1. Materials

The membranes were prepared using cassava starch (Tecnas S.A., Cali, Colombia), sodium hydroxide, glycerol (food grade), and glacial acetic acid (Merck, Burlington, MA, USA). All chemicals were of analytical degree and were used without further purification unless otherwise indicated. Gelatin’s chicken extraction was reported elsewhere [36]. All the chemical reagents had analytical, and no further purification was performed unless stated otherwise.

### 2.2. Methods

#### 2.2.1. Preparation of the Sample of Chicken Gelatin

Chicken gelatin (Figure 1) was prepared according to the previously reported work [36]. The cassava starch suspension was initially heated at 85 °C with constant stirring for one hour. Later, the cassava starch was mixed with glycerol at room temperature for 10 min. A weight ratio of glycerol: starch of 25:100 was used. Subsequently, the suspension was heated to 80 °C for 15 min, cooled to 60 °C, and slowly added the amount of gelatin. Finally, the rest was sonicated at 60 °C for 50 min using an ultrasonic bath (Branson, Madrid, Spain). The suspensions in plastic molds were environmentally cured for 24 h and placed for another 24 h in an oven at 40 ± 0.2 °C.

#### 2.2.2. Synthesis of Cassava Starch/Gelatin Composite Membranes

Membrane synthesis followed Podshivalov’s methodology [37], as indicated elsewhere [36]. The same five formulations and procedures were used for this batch, as shown in Table 1:

#### 2.2.3. Membrane Characterization

The physicochemical analysis of the composites used FT-IR spectroscopy, XRD technique, thermal analysis, and morphology studies effects under gelatin incorporation to the starch.

##### Thermal Analysis

Thermal stability and transitions used a TGA/DSC 2 STAR System instrument (Mettler Toledo, Columbus, OH, USA). Heating program: from 25 °C to 900 °C; heating rate: 20 °C/min; nitrogen flow: 60 mL/min. Weight loss was shown as a temperature function from a single sample run. Heat flow was shown as a function of temperature respectively.

##### Differential Scanning Calorimetry (DSC)

DSC was used for gelatinization temperature and enthalpy (*ΔH_m_*) measurements with a DSC1 Star system/500 (Mettler Toledo, Schwerzenbach, Switzerland) at the first heating/cooling cycle from a single run of a sample. Heating cycle: from −25 °C to 300 °C; heating rate: 10 °C/min; nitrogen flow: 60 mL/min. Data analysis used the Star^e^ Software version 12.10, (Schwerzenbach, Switzerland).

##### Functional Group Characterization of the Membranes

Composite functional groups analysis used an FT-IR instrument in the ATR mode with a diamond tip (Shimadzu, Kyoto, Japan). Scans from 500 and 4000 cm^−1^.

##### Morphology Analysis

A JEOL, JCM 50,000 (Tokyo, Japan) scanning electron microscope (SEM) was employed for the morphology study of the composite membrane cross-section analysis. The samples were covered with a gold layer. The surface membranes were analyzed at 500× and 2000× magnification. A voltage of 10 kV was applied.

##### X-ray Diffraction (XRD)

X-ray diffractometry (XRD) used a PANalytical X’ Pert PRO diffractometer (Malvern Panalytical); irradiation source: CuKα radiation (λ = 0.154 nm). The 2θ range: 5–60°.

#### 2.2.4. In Vivo Biocompatibility Tests

Nine male Wistar rats, four months old and weighing approximately 360 g, were taken from the LABBIO laboratory of the Universidad del Valle in Cali, Colombia, and randomly distributed into three groups for implantation at 30, 60, and 90 days.

Animals’ anesthesia was applied intramuscularly using Ketamine 70 mg/kg (Blaskov Laboratory, Bogotá, Colombia) and Xylazine 30 mg/kg (ERMA Laboratories, Celta, Colombia). Trichotomy of the dorsal surface, disinfection with isodine^®^ solution (Sanfer Laboratory, Bogotá, Colombia), and infiltration with 2% Lidocaine with epinephrine (Lidocaine, Newstetic, Guarne, Colombia) were performed. Five pockets 1 cm long by 2 cm deep were made on the right dorsal surface, and the experimental formulations consisting of membranes 0.5 mm wide by 10 mm long were placed. Each pocket was sutured with absorbable 4 zero suture (Vicryl, ETHICON, Johnson and Johnson, New Brunswick, NJ, USA); Topical gentamicin (Gentamicin 1%, Procaps, Cali, Valle del Cauca, Colombia), and intramuscular Diclofenac 75 mg (La Sante, Bogotá, Colombia) were applied, food and water were provided ad libitum.

There were no complications or deaths in biomodels due to the procedures. Once the implantation period was completed, the biomodels were euthanized using intraperitoneal sodium pentobarbital/sodium diphenylhydantoin (0.3 mL/biomodel kg) (Euthanex, INVET Laboratory, Cota, Colombia).

The samples were recovered and fixed with buffered formalin for 48 h. Subsequently, they were washed with phosphate buffer (PBS) for 10 min in 3 exchanges and processed for histological techniques with the Autotechnicon Tissue ProcessorTM equipment (Leica Microsystems, Mannheim, Germany).

Paraffin blocks were formed with the samples using the Thermo ScientificTM HistoplastTM equipment (Thermo Fisher Scientific, Walthman, MA, USA), cuts were made at 6 µm with the Leica RM2125 RTS microtome (Leica Microsystem, Mannheim, Germany), and stains were made with Hematoxylin-Eosin and Masson’s Trichrome techniques.

Photomicrographs were taken using a Leica DM750 optical microscope and a Leica DFC 295 camera. Images were processed with Leica Application Suite version 4.12.0 software (Leica Microsystem, Mannheim, Germany).

The procedures were carried out using the recommendations of the ARRIVE guide (Animal Research: Reporting of In Vivo Experiments), and the ethical review was carried out by the Animal Ethics Review Committee of the Universidad del Valle (Cali, Colombia) through Resolution No. CEAS 012 of 2019.

## 3. Results and Discussion

### 3.1. FT-IR Spectroscopy Analysis of the Cassava Starch/Gelatin Membranes

We can observe the FT-IR spectrum for all the composite membranes in Figure 1. The peak at 3320 for T1–T5 can be attributed to the OH groups from starch and adsorbed water, which is broader for T1 (100%CS) and T2 (25%G/75%CS) (higher starch content). However, with the increase in gelatin content (from T2–T5), the peak is sharper and more intense due to a higher amide content [38]. All the expected bands for cassava starch are present in Figure 1. The –C–H vibration stretching bands are observed at 2924 cm^−1^ in T1, but it becomes sharper and more intense with the increasing gelatin content indicating a higher C–H content (from T2 to T5). The bands associated with in-plane bending vibrations for the CH_2_ and C–OH groups are shown at 1422 and 1337 cm^−1^, respectively. However, these bands are evident for T1 but less clear with the decreasing amount of starch (From T2–T5). Another peak for T1 due to the antisymmetric bridge of the C–O–C groups rises at 1153 cm^−1^ [31].

It was interesting that for T2–T5, with the increase in protein content, the amide I stretching bands increased and shifted from 1654 to 1645 cm^−1^, while the peak at 1744 cm^−1^ indicated the gelatin’s −C=O group presence for T2–T5, also increasing with the gelatin content, except for T4, where it was overlapped by the amide I band, probably due to a high amount of hydrogen bonds between starch and gelatin chains [39]. Furthermore, the sharper band for the −OH group amide band shifting is probably due to the formation of hydrogen bonds between the groups C=O y N−H with the group −OH. The increased gelatin content in the membranes (from T2, 25%G/75%CS to T5, 100% G) also decreased the C–O–C bands in starch at 999 cm^−1^. In the case of T3, the small peak at 1026cm^−1^ was probably due to the 50:50 content of starch: gelatin introducing hydrogen bonds [40]. T5 presented a band at 1744 cm^−1^ from C=O groups of amides-I and a band at 1548 cm^−1^ for amides−II from the C−N bonds [41].

### 3.2. Thermal Analysis of Cassava Starch/Gelatin Membranes

Figure 2 shows the thermal degradation curves for T1–T5, which mainly presented three degradation steps (except T4 with only two). Generally, in all the membranes, the presence of water was observed between 50–150 °C. The second weight loss step began at 250–270 °C, corresponding to some volatile organic compounds and changes in the helix structures. Finally, the third step of weight loss (300–338 °C) was attributed to the decomposition of the polyamide and glycosidic break bonds (polysaccharides and protein) in the membrane.

Concerning the formulations containing gelatin/cassava starch was observed that the formulations T3 (50%G/50%CS) and T4 (75%G/25%CS) above 350 °C have an effective stabilization (lower mass loss) compared with the membranes T1 (100%CS), T2 (25%G/75%CS), and T5 (100%G) (Figure 2A). This observation is probably due because, during the membrane formation, a swelling process breaks unions between amylopectin units present in the cassava starch. The amylose is crosslinked with the amylopectin units released during the thickening [42]. Therefore, an increase in the flexibility of the material was observed due to the union between the main components. Additionally, amylose can act as a “glue” between the amylopectin lamellae, which has a similar effect. This effect is consistent with previously observed results [43]. Figure 2B shifted the third degradation from 320 °C for T1 (with a sharp and intense peak) to 306 °C for T4. The increase in the gelatin content affects the polymer chain order, and more crosslinked chains are formed with the temperature increase, which also supports the stabilization effect with the increased residual mass at 800 °C (~15 wt.%) observed in the TGA plot (Figure 2A).

DSC enables the determination of mesomorphic transitions, melting and crystallization temperatures, entropy and enthalpy changes, and the characterization of glass transition temperatures [44]. For all the formulations, it can be observed that there is an initial gradual drop in the crystallinity index between the room temperature and the peak endothermic in the DSC (Figure 3). Therefore, these plots represent a crystallinity loss occurring at a gelatinization endothermic temperature [45]. When samples are subjected to high temperatures, the energy absorbed by the components modifies their crystalline structure leading to the rearrangement of the amylopectin and amylose helixes [45]. This interpretation is consistent with experiments with potato starch [46]. For this type of membrane, it is expected to mention the enthalpy of gelatinization instead of the enthalpy of dehydration because it is an intrinsic measure of the material’s hydration level [47]. Therefore, this enthalpy is related to the molecular order of the components, which are modified when subjected to high temperatures leading to a change in the material’s crystallinity [48].

The gelatinization heats measured for samples T1, T2, T3, T4, and T5 were 211.24, 226.90, 195.67, 196.40, and 231.31 J/g, respectively. This means that T2, T3, and T4 composites need less energy to dissociate the amylose and amylopectin helix due to the lower starch content causing less rigidity [42] which is consistent with experiments containing gelatin within biopolymer or polymeric matrices [49].

### 3.3. Scanning Electron Microscopy (SEM) of Cassava Starch/Gelatin Membranes

It is interesting to observe in Figure 4 the morphological analysis of the cross-section of starch/gelatin composite membranes. For T1 (100%CS), a relatively compact polysaccharide chain structure was observed in the cross-section (Figure 4A) or on the surface (Figure 4B), but some accumulation was also observed (arrow Figure 4A). On the other hand, with the gelatin increasing (from 25 to 75 wt.% for T2–T4, respectively), a rough, heterogeneous, and discontinuous appearance with cracks and pores (Figure 4B,J) as marked with the arrows, due to the gelatin flexibility polymer phase-separation [50]. It has been previously demonstrated that low protein-membrane content produced poor polymer interactions, while high-globular-protein content decreased water activity, making more flexible and brittle membranes [50]. T2–T4 membranes presented rougher morphologies in the cross-sections with macro porosity, especially for T4 (Figure 4G,H) due to higher protein content (which increased the hydrophobic character of the composites).

Furthermore, the porosity of T2–T4 demonstrates composite microstructure differences due to the different molecular arrangements between amylose and amylopectin [49]. T5 was also compact and had less cracking due to higher compatibility between the protein molecules, which increased the hydrogen bonds [51,52]. However, there were differences in the appearance of the starch/gelatin membranes. With the increasing gelatin content (especially for T3 and T4), a rougher and more heterogeneous structure was observed, similar to previous observations for starch/bovine type b gelatin mixtures [53].

### 3.4. X-ray Diffraction (XRD)

XRD studies are related to the chemical composition, phases present, and crystallinity of the composite structures [54]. Figure 5 shows a diffractogram of our different mixed formulations.

According to Figure 5, the crystallization for T1–T5 presented a similar amorphous behavior. A prominent broad peak at 20° with a shoulder for T1–T5 corresponded to a B-type pattern from a high amylose cassava starch [55]. For T2 and T3, with a high content of starch mixed with gelatin, only a tiny amount of amylose reacted with gelatin, generating a crystalline pattern similar to T1 due to the recrystallization of amylose molecules after the starch gelatinization [56]. The peak at 22° and 23°, observable only for T3 and T5 as a shoulder, could be due to the specific pattern for collagen, the main protein constituent of the gelatin [57]. Usually, a peak centered at 8° is related to the triple helix structure of collagen [58], which was not observable here, indicating that the triple helix structure was disrupted by the hydrogen bonding with the starch polymer chains [59,60]. Moreover, it is essential to observe the decrease in the peak intensity centered at 20° from T1 to T3 due to the lost crystallinity of the membranes and the increased phase separation between the polymeric chains, a consistent result with the SEM and DSC analysis.

### 3.5. Biological Tests

#### 3.5.1. In Vivo Biocompatibility Assessment

Once the euthanasia was carried out and under the UNE EN ISO 10993-6:2017 standard (Biological evaluation of medical devices—Part 6: Tests for local effects after implantation), a macroscopic inspection of the intervened tissues was carried out. In all cases, hair recovery occurred. In addition, healthy tissues were observed when performing skin trichotomy in the dorsal area of the biomodels. On the other hand, when dissecting the skin to retrieve the samples from the implantation area, the skin was completely healed, with no signs of inflammation or infection.

Figure 6 corresponds to the macroscopic inspection of a biomodel after 30 days of implantation. The figure shows the conditions in which the tissue was found (Figure 6A,B). The areas where the materials were implanted are not observable and are confused with the untreated tissue (Figure 6C).

#### 3.5.2. Histology Results of T1 (100%CS)

The T1 sample was characterized by presenting a very rapid reabsorption/degradation process, with the presence of large blood vessels in the implantation area. Figure 7 corresponds to the histological studies of formulation T1 (100%CS).

It is observed that 30 days after implantation, the material remains in the implantation area (Figure 7A). The small sheets of material in a concentric arrangement are immersed in a connective tissue for healing made up of type I collagen with the presence of numerous blood vessels (Figure 7B). At 60 days, the amount of remnant material is insignificant, with only a few remains found in a single biomodel (Figure 7C). For the 90 days, it was impossible to observe the presence of the remaining T1 material in any of the three intervened biomodels, evidencing total reabsorption of the material.

Macroscopic and microscopic observations show that the material behaved as biocompatible and bioresorbable/biodegradable. The healing process was carried out without the fibrous encapsulation reported as typical of an observable foreign body response when biomaterials are implanted subdermally. For T1 (only starch), reabsorption seems to have been completed before 60 days, which can be explained by the fact that starch is an entirely resorbable material [61] being degraded by amylases [62].

#### 3.5.3. Histology Results of T2 (25%G/75%CS)

T2 presents more stability than T1, remaining stable for at least 60 days due to the presence of 25 wt.% of gelatin. After 30 days, fragments of the material are immersed in connective tissue for healing made up of type I collagen (Figure 8A,B). At 60 days, the samples were still histologically observable (Figure 8C,D). However, after 90 days, the material was degraded entirely/resorbed for the tissue.

Material T2 differs from T1 in that 25 wt.% gelatin has been added to its composition. This amount was sufficient to allow excellent stability to the compound, making it observable for up to 60 days. Besides, as observed for T1, the healing process was carried out without generating the foreign body reaction.

#### 3.5.4. Histology Results of T3 (50%G/50%CS)

The macroscopic examination of the areas implanted with the T3 formulation showed biocompatibility reflected in the tissues’ scarring process, leading to hair recovery and a standard appearance without fistulas or the presence of purulent exudate.

Microscopic examination indicates a healing process. In this process, the implantation area was surrounded by a fibrous capsule with an inflammatory cell infiltration (Figure 9A). This capsule comprises type I collagen fibers, showing that the reabsorption process was normal. However, after 60 days, this capsule completely disappears as the inflammatory infiltrates (Figure 9C), although some material particles remain observable (Figure 9D). After 90 days, the presence of material or the implantation zone is no longer noticeable, evidencing complete resorption and recovery of the cellular architecture.

The healing process results show that adding 50% gelatin makes the material more stable, with membranes observable after 60 days. The healing is carried out by reacting to a foreign body with fibrous encapsulation of the material; however, after 90 days, neither the fibrous capsule nor the material is observable.

#### 3.5.5. Histology Results of T4 (75%G/25%CS)

Like the previous results, the macroscopic inspection of the intervened areas showed initial healing with normal parameters such as hair recovery and the absence of signs of inflammation or infection.

At the microscopic level, fragments of the material immersed in healing connective tissue with the presence of inflammatory cells are observed after 30 days (Figure 10A); at 60 days, the inflammatory infiltrate is mild, and the implanted material is barely observable. At 90 days, it is not possible to observe the material. Through Masson’s trichrome staining, it is possible to follow some type I collagen fibers forming the connective tissue of cicatrization.

It is important to remember that the T4 material has a 75 wt.% gelatin content composition. After 30 days of implantation (Figure 10A), the material is immersed in an inflammatory infiltrate that facilitates the degradation, which remains perceptible after 60 days (Figure 10B). However, a complete degradation/reabsorption is evident after 90 days, when a fully recovered tissue is found (Figure 10C,D).

#### 3.5.6. Histology Results of T5 (100%G)

The implantation areas of the T5 material presented a macroscopic image of healing very similar to that already described for the other formulations. In the macroscopic aspect, after 30 days, the material is immersed in a connective healing tissue (Figure 11A) composed of type I collagen fibers (Figure 11B). After 60 days, most of the material appears to have been resorbed/degraded but with observable fragments (oval area Figure 11C). For 90 days, it is still possible to observe pieces of the material (circle area in Figure 11D).

The T5 material is no longer made up of starch but of gelatin. It was observed that at 30 days, the material incited a reaction by inflammatory cells, and the material was highly fragmented. At 60 days, a portion of the implanted material is observable, surrounded by a fibrous capsule. After 90 days, small fragments of the material in the process of degradation/reabsorption remain observable.

When materials are implanted, there is expected to be a foreign body reaction in the final stage of healing with a fibrous capsule surrounding the material [63]. Capsule formation is how the organelle limits foreign material, while hydrolytic or enzymatic degradation of the material occurs or is reabsorbed by phagocytic-type inflammatory cells [64].

The foreign body reaction with fibrous encapsulation will continue the recovery process of the tissues affected by the surgical preparation. The foreign body reaction is expected to continue if there is remaining material. At the same time, the situation caused by the material’s presence is resolved.

Fibrous encapsulation occurs in the resolution stage of the foreign body reaction. In this phase, there is an increase in the production of the TGFβ factor by macrophages and fibroblast-like cells. As a result, type I and Type III collagen bundles are produced [65].

The presence of this fibrous tissue was evident in formulations containing a percentage of gelatin greater than 50% (T3, T4, and T5) but not in formulations T1 (100% starch), or T2 (75% starch/25% gelatin), because starch is entirely degraded by amylases avoiding the foreign body response.

Gelatin is considered a biocompatible and fast-reabsorbing material [66]; however, in applications of gelatin hydrogels made with different types of gelatin and techniques, it was found that in vivo degradation occurs by enzymatic digestion and is affected by the gelling manufacturing technique [67]. In this investigation, it was found that the compounds containing between 25 wt.% and 75 wt.% of gelatin presented complete resorption before 90 days with the recovery of standard tissue architecture. On the other hand, T5 formulation at 90 days exhibited small fragments surrounded by a fibrous capsule. In experimental implantations, it was found that gelatin sponges could cause high fibrosis and foreign body reaction with progressive reabsorption of the material [68,69].

Gelatin is obtained from various types of collagens; however, the degradability of collagen is different from that of gelatin. It has been found that gelatin degrades/reabsorbs faster than collagen because it can induce a foreign body reaction with many giant cells that would phagocytize the implanted material more efficiently [70]. Histological images of formulations T4 and T5 with high gelatin content show that the material incited a foreign body reaction by infiltrating inflammatory cells with rapid degradation, leaving only a few small fragments for the T5 formulation in 90 days.

This research showed that the material obtained has potential use as a dressing for skin lesions due to trauma or burns. In bone regeneration applications, however, this work has the limitation that the animal experimentation was performed using a subdermal model, which is valid for the preliminary results obtained. For that reason, it is suggested to perform other types of research in the future to evaluate the usefulness of regenerative bone techniques or their application to the skin. For bone regeneration, experimental models should allow qualitatively and quantitatively results for skin dressing, as proposed by Gutierrez and collaborators [71].

We cannot directly compare our system to commercial products since commercial membranes for tissue engineering are made basically from collagen. However, our previous work [36] determined that the obtained cassava starch/gelatin composite films had good mechanical properties comparable to the reported values of commercial polyethylene films. With the characterization results for the composites, we brought an optimized formulation for the cassava starch/gelatin-based films in a 53/47 ratio, plasticized with glycerol using the casting method that would meet the expectations of the model polyethylene film for food-packaging applications. Moreover, the FT-IR and the thermal analysis are according to this previous work, as stated in the manuscript. For tissue engineering applications, a porcine collagen membrane with high porosity, fluid absorption, and resorption capacity, stretchable in all directions with fibrous characteristics, is a highly-resorbable material, as the manufacturer stated (InterCollagen Guide, SigmaGraft, Inc., Fullerton, CA, USA). As we published earlier, we performed the physical-chemical characterization of this collagen membrane [72]. The most crucial result in the present study is that the T1–T5 membranes present interconnected macro porosity as observed in the SEM results but are more stable under degradation in body conditions than the InterCollagen Guide commercial product, without offering an exacerbated response. These results demonstrate that the cassava starch/chicken gelatin membranes obtained here are good candidates for long-term tissue engineering applications that can be studied in the future, such as bone tissue regeneration.

## 4. Conclusions

The physical-chemical characterization of the membranes demonstrated that the method was affordable for membrane synthesis. The FT-IR analysis revealed the presence of the leading bands of starch and gelatin polymers. Besides, for T2–T5, with the increase in protein content, the amide I stretching bands increased and shifted from 1654 to 1645 cm^−1^, while the peak at 1744cm^−1^ indicated the gelatin’s −C=O group presence for T2–T5, also increasing with the gelatin content, demonstrating the existence and interactions between starch and gelatin polymers. The thermal behavior of the membranes showed an internal crosslinking between the protein chains as the starch content decreased. We found that morphologies and crystallinities were influenced by the amylose content inside the starch, as evidenced by the broad peak at 20° on XRD. The increasing amount of gelatin strongly influenced the morphology. The cross-section and the surface structures demonstrated macro porosity and roughness increasing with the gelatin presence in the composites in a gelatin-amount-dependent manner. T4 presented the more heterogeneous structure between the composites.

The five implanted formulations showed to be biocompatible and biodegradable. In the case of formulations with high starch content (T1 and T2), the degradation is faster because the material is enzymatically (amylase) degraded due to high starch content. According to the histology analysis, the T3 formulation (50% starch/50% gelatin) was more stable under the degradation/resorption process than T1 and T2. With the increase in gelatin, higher resistance to degradation was observed. However, most of these membranes at 60 days (T1, T2, and T3) disappear as a sign of the degradation/resorption process.

T5 induced a foreign body reaction with the presence of cells. However, it undergoes a progressive degradation/resorption without an allergic response, demonstrating the materials’ in vivo biocompatibility. On the other hand, the fragments of the T4 and T5 formulations are more stable, and degradation is observed only at 90 days. All these results demonstrate the preliminary biocompatibility of the composites and the potential for biomedical applications.

## Data Availability

Samples are available under request to the corresponding author.

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
