# Peer review of "Histological Evaluation of Cassava Starch/Chicken Gelatin Membranes"

_polymers, 2022, doi:10.3390/polym14183849_

Round 1

Reviewer 1 Report

Dear Authors,

This manuscript covers a lot of research and is written quite well. However, in order to improve its quality, the manuscript should be completed as indicated in this review.

Detailed comments below:

Line 62: Starch is also a common polymer found in grains of cereals, rice, etc. Complete this information or add a generalization.

Line 89: Add more information about these tissues. Give examples of hard tissues in which these biocomposites can be used. More emphasize the scientific aspect of your research.

I also believe that you should precisely indicate the purpose of this article.

Line 103: What was the percentage of glycerol added? In table 1 you only indicate the addition of starch and gelatin.

Line 112: Nevertheless, it is worth describing (briefly) how the foil was produced in your laboratory conditions.

Line 218: Write if these values ​​are satisfactory. Are the obtained results close or far compared to other commercial materials?

These questions basically apply to all the results obtained. Review all research results for this.

Figure 4: Indicate (e.g. with arrows, circle ... etc.) the areas to be described on the selected images.

Line 430: Conclusions are well written but I noticed some shortcomings in relation to e.g. FTiR analysis. Go through the article point by point and complete your conclusions.

Author Response

The authors are deeply thankful for the constructive criticism for improving the manuscript. All the suggestions are answered point by point. We hope all the recommendations were well attended to. 

Reviewer 1

This manuscript covers a lot of research and is written quite well. However, in order to improve its quality, the manuscript should be completed as indicated in this review.

 Detailed comments below:

Line 62: Starch is also a common polymer found in grains of cereals, rice, etc. Complete this information or add a generalization.

R// We appreciate the valuable suggestion. The information was completed in lines 62-63.

Line 89: Add more information about these tissues. Give examples of hard tissues in which these biocomposites can be used. More emphasize the scientific aspect of your research. I also believe that you should precisely indicate the purpose of this article.

R// We appreciate the valuable corrections. In the paper, the information can be found between lines 91-114

In this regard, generating different mixtures between gelatin and various components would improve the physical conditions in a biological environment. The improvement would involve the formation of scaffolds for skin lesions or as barriers in processes where it is required to promote bone healing, such as the guided bone regeneration technique. The skin is a vital organ with many functions to protect the tissues; when part of it is lost due to accidents or burns, it is essential to protect the injured tissue by employing scaffolds to prevent water loss and infection. The scaffolds can be natural or synthetic and must comply with the properties of being biocompatible, stable for the healing duration, absorbent of tissue exudates, and ideally with the ability to bio-integrated allowing the closure of the wound properly and be bacteriostatic [34].

In guided bone regeneration techniques, the aim is to stimulate tissue neoformation by using bone substitutes and placing a membrane between the graft material and the soft tissue to act as a barrier that prevents the smooth tissue cells from colonizing the graft site. This barrier must be made of a biocompatible material, one of the most commonly used collagen [35]. 

We published the synthesis and physicochemical characterization of five different formulations of membrane cassava starch and chicken gelatin composites for food packaging applications [36]. However, despite the recorded applications of cassava starch with gelatin membranes for food packaging, these membranes' biocompatibility has not been previously reported. Therefore, we report the study of in vivo biocompatibility during 90 days of implantation of five different formulations of cassava starch and chicken gelatin biocomposites. The results show that this type of membrane is beneficial for regenerating subdermal tissue, thanks to its rapid reabsorption in the tissues without generating an abnormal immune response in the host.

Line 103: What was the percentage of glycerol added? In table 1 you only indicate the addition of starch and gelatin.

R// We appreciate the suggestion. The information was added in line 128.

Line 112: Nevertheless, it is worth describing (briefly) how the foil was produced in your laboratory conditions.

R// We appreciate the suggestion. The preparation of films was described between lines 124 to 132.

Line 218: Write if these values ​​are satisfactory. Are the obtained results close or far compared to other commercial materials? These questions basically apply to all the results obtained. Review all research results for this.

R// We appreciate the observation. We added the following information between lines 488-514:

This research showed that the material obtained has potential use as a dressing for skin lesions due to trauma or burns. In bone regeneration applications, however, this work has the limitation that the animal experimentation was performed using a subdermal model, which is valid for the preliminary results obtained. For that reason, it is suggested to perform other types of research in the future to evaluate the usefulness of regenerative bone techniques or their application to the skin. For bone regeneration, experimental models should allow qualitatively and quantitatively results for skin dressing, as proposed by Gutierrez and collaborators [71].

We cannot directly compare our system to commercial products since commercial membranes for tissue engineering are made basically from collagen. However, our previous work [36] determined that the obtained cassava starch/gelatin composite films had good mechanical properties comparable to the reported values of commercial polyethylene films. With the characterization results for the composites, we brought an optimized formulation for the cassava starch/gelatin-based films in a 53/47 ratio, plasticized with glycerol using the casting method that would meet the expectations of the model polyethylene film for food-packaging applications. Also, the FT-IR and the thermal analysis are according to this previous work, as stated in the manuscript. For tissue engineering applications, a porcine collagen membrane with high porosity, fluid absorption, and resorption capacity, stretchable in all directions with fibrous characteristics, is a highly-resorbable material, as the manufacturer stated (InterCollagen Guide, SigmaGraft, Inc., Fullerton, CA, USA). As we published earlier, we performed the physical-chemical characterization of this collagen membrane [72]. The most crucial result in the present study is that the T1-T5 membranes present interconnected macro porosity as observed in the SEM results but are more stable under degradation in body conditions than the InterCollagen Guide commercial product, without offering an exacerbated response. These results demonstrate that the cassava starch/chicken gelatin membranes obtained here are good candidates for long-term tissue engineering applications that can be studied in the future, such as bone tissue regeneration.

Figure 4: Indicate (e.g. with arrows, circle ... etc.) the areas to be described on the selected images.

R// We appreciate the observation. New information was added between lines 293-312 (section 3.3)

It is interesting to observe in Figure 4 the morphological analysis of the cross-section of starch/gelatin composite membranes. For T1 (100%CS), a relatively compact polysaccharide chain structure was observed in the cross-section (Figure 4A) or on the surface (Figure 4B), but some accumulation was also observed (arrow Figure 4A). On the other hand, with the gelatin increasing (from 25 to 75 wt.% for T2-T4, respectively), a rough, heterogeneous, and discontinuous appearance with cracks and pores (Figures 4B – 4J) as marked with the arrows, due to the gelatin flexibility polymer phase-separation [49]. It has been previously demonstrated that low protein-membrane content produced poor polymer interactions, while high-globular-protein content decreased water activity, making more flexible and brittle membranes [49]. T2 - T4 membranes presented rougher morphologies in the cross-sections with macro porosity, especially for T4 (Figures 4G and 4H) due to higher protein content (which increased the hydrophobic character of the composites).

Furthermore, the porosity of T2-T4 demonstrates composite microstructure differences due to the different molecular arrangements between amylose and amylopectin [50]. T5 was also compact and had less cracking due to higher compatibility between the protein molecules, which increased the hydrogen bonds [51,52]. However, there were differences in the appearance of the starch/gelatin membranes. With the increasing gelatin content (especially for T3 and T4), a rougher and more heterogeneous structure was observed, similar to previous observations for starch/bovine type b gelatin mixtures [53].

Line 430: Conclusions are well written but I noticed some shortcomings in relation to e.g. FTiR analysis. Go through the article point by point and complete your conclusions.

R// We appreciate the reviewer's comment. The information added can be found between lines 517-543.

The physical-chemical characterization of the membranes demonstrated that the method was affordable for membrane synthesis. The FT-IR analysis revealed the presence of the leading bands of starch and gelatin polymers. Besides, for T2-T5, with the increase in protein content, the amide I stretching bands increased and shifted from 1654 to 1645 cm-1, while the peak at 1744cm-1 indicated the gelatin's -C=O group presence for T2-T5, also increasing with the gelatin content, demonstrating the existence and interactions between starch and gelatin polymers. The thermal behavior of the membranes showed an internal crosslinking between the protein chains as the starch content decreased. We found that morphologies and crystallinities were influenced by the amylose content inside the starch, as evidenced by the broad peak at 20° on XRD. The increasing amount of gelatin strongly influenced the morphology. The cross-section and the surface structures demonstrated macro porosity and roughness increasing with the gelatin presence in the composites in a gelatin-amount-dependent manner. T4 presented the more heterogeneous structure between the composites.

The five implanted formulations showed to be biocompatible and biodegradable. In the case of formulations with high starch content (T1 and T2), the degradation is faster because the material is enzymatically (amylase) degraded due to high starch content. According to the histology analysis, the T3 formulation (50% starch/50% gelatin) was more stable under the degradation/resorption process than T1 and T2. With the increase in gelatin, higher resistance to degradation was observed. However, most of these membranes at 60 days (T1, T2, and T3) disappear as a sign of the degradation/resorption process.

T5 induced a foreign body reaction with the presence of cells. However, it undergoes a progressive degradation/resorption without an allergic response, demonstrating the materials' in vivo biocompatibility. On the other hand, the fragments of the T4 and T5 formulations are more stable, and degradation is observed only at 90 days. All these results demonstrate the preliminary biocompatibility of the composites and the potential for biomedical applications.

Reviewer 2 Report

Dear Editor, dear Authors, Carlos Humberto Valencia-Llano et al., submitted a paper on the in vivo assessment of the biocompatibility of cassava starch/gelatin films composites after subdermal implantations in Wistar rats as a biomodels at 30, 29, 60, and 90 days. The authors have characterized the produced films composites using several techniques such as FTIR, TGA, DSC, XRD and SEM. They have also demonstrated that the five implanted formulations with 0, 25, 50, 75 and 100 % gelatin are biocompatible and biodegradable with faster degradation observed in formulations of higher starch content. I believe that the manuscript is of interest, and well written. The results are well supported with experimental evidence and the manuscript can be accepted for publication in. However, I have some minor comments to the authors.

Comments to the author:

-        Page 4 lines 181 to 183 the authors wrote “bands associated with bending vibration of CH2 and C-OH groups are evident at 1422 cm-1 for T1 but less clear with the decreasing amount of starch (From T2-T5).” The authors should clarify here, is 1422 cm-1 is for the vibration of C-OH or CH2. ?

-          Page 5 lines 192 and 193 “with the CH-O-CH2 groups of starch from 999 cm-1 to 1040 cm-1.” What the authors mean by CH-O-CH2 groups of starch, is the band for C-O-C or for CH of starch please revise and clarify.

-          I suggest that the authors include the chemical structures of starch and gelatin in the manuscript to explain further the FTIR results.

 Sincerely Yours,

Author Response

We hope that all the requirements have been well attended to. There are answers point by point to all the recommendations. 

Reviewer 2

Dear Editor, dear Authors, Carlos Humberto Valencia-Llano et al., submitted a paper on the in vivo assessment of the biocompatibility of cassava starch/gelatin films composites after subdermal implantations in Wistar rats as a biomodels at 30, 29, 60, and 90 days. The authors have characterized the produced films composites using several techniques such as FTIR, TGA, DSC, XRD and SEM. They have also demonstrated that the five implanted formulations with 0, 25, 50, 75 and 100 % gelatin are biocompatible and biodegradable with faster degradation observed in formulations of higher starch content. I believe that the manuscript is of interest, and well written. The results are well supported with experimental evidence and the manuscript can be accepted for publication in. However, I have some minor comments to the authors.

R// We appreciate the positive comments from the reviewer.

 Comments to the author:

-        Page 4 lines 181 to 183 the authors wrote "bands associated with bending vibration of CH2 and C-OH groups are evident at 1422 cm-1 for T1 but less clear with the decreasing amount of starch (From T2-T5)." The authors should clarify here, is 1422 cm-1 is for the vibration of C-OH or CH2. ?

R// We appreciate the reviewer's comment. The corrected paragraph can be found on lines 211-215.

"The bands associated with in-plane bending vibrations for the CH2 and C-OH groups are shown at 1422 and 1337 cm-1, respectively. However, these bands are evident for T1 but less clear with the decreasing amount of starch (From T2-T5). Another peak for T1 due to the antisymmetric bridge of the C-O-C groups rises at 1153 cm-1".

-          Page 5 lines 192 and 193 "with the CH-O-CH2 groups of starch from 999 cm-1 to 1040 cm-1." What the authors mean by CH-O-CH2 groups of starch, is the band for C-O-C or for CH of starch please revise and clarify.

R// We appreciate the reviewer's comment. This appreciation can be found on lines 223-227.

Furthermore, the sharper band for the -OH group amide band shifting is probably due to the formation of hydrogen bonds between the groups C=O y N-H with the group -OH. The increased gelatin content in the membranes (from T2, 25%G/75%CS to T5, 100% G) also decreased the C-O-C bands in starch at 999 cm-1.

-          I suggest that the authors include the chemical structures of starch and gelatin in the manuscript to explain further the FTIR results.

R// We appreciate the reviewer's comment. The new structures can be found between lines 133-135.

Reviewer 3 Report

Manuscript ID: polymers-1894659
This manuscript, Histological Evaluation of Cassava Starch/Chicken Gelatin Membranes, is an interesting paper focusing on novel cassava starch/gelatin films and their properties as well as behavior in a living organism. The manuscript is well-thought, contains very valuable results and good scientific discussion, although, there are several points that should be improved: [1] Please standardize the nomenclature throughout the work (scaffolds, membranes, films). [2] In the Introduction please emphasize the purpose of this work in more detail. [3] In the Materials please describe the properties of the reagents (such as. e.g. purity). [4] I strongly recommend that the authors should change the specimen's nomenclature, e.g. T1 -> 100%CS, T2 -> 25%G/75%CS etc. This will make reading easier for the potential reader. [5] The synthesis should be briefly described. [6] In Fig. 1 please add descriptions of peaks. [7] In Figs. 2 and 3 please add the number of repetitions (n=x). [8] In Fig. 4 please add magnification in the text. [9] In The Discussion please add a description of limitations and future plans. [10] The Conclusions should be shortened and contain only the most important conclusions from the research. [11] There is no description of the Author's Contributions.

Author Response

We hope that all the recommendations and the suggestions from the reviewer have been satisfied. We answered point by point, as you can see from the attached letter and the second version of the manuscript. 

Round 2

Reviewer 1 Report

Dear authors,

I accept all corrections.